# Treatment of Scrap Tire for Rubber and Carbon Black Recovery

**Alaa Sultan Abdulrahman and Fawzi Habeeb Jabrail ***

Polymer Research Laboratory, Department of Chemistry, College of Science, University of Mosul,
Mosul 41001, Iraq; alaa.sultan.007@gmail.com
* Correspondence: fawzijabrail@uomosul.edu.iq; Tel.: +964-77-0333-6282

**Abstract:** In this study, a chemical dissolution treatment was used to recover rubber and carbon black (CB) from truck tire scrap, with gas oil acting as the solvent and 4-Hydroxy-TEMPO acting as the catalyst for the chemical reactions. Montmorillonite clay was used to separate the rubber solution from the CB and the other non-dissolved tire additives. The recovered rubber and CB were characterized together with the original scrap tire sample by XRD, SEM, BET and thermal analysis, as well as FTIR and $^1$H NMR spectral analyses. Characterization of the chemical structure of the recovered rubber showed that the main functional groups of styrene−butadiene rubber blend with natural rubber. The thermal behavior and crystalline structure of the recovered rubber, as well as its morphological images, showed that the properties of the rubber sample were acceptable and similar to natural rubber. The recovered CB characterizations showed that the sample after pyrolysis was a highly crystalline nanocomposite structure with a high specific surface area and scattered pores.

**Keywords:** recycling of scrap tire; recovery of rubber; recovery of carbon black; pyrolysis; montmorillonite; gas oil solvent





## 1. Introduction

Treatments for accumulated tire waste have become an urgent necessity. Throughout the world huge quantities of tires wastes are creating significant environmental pollution problems, affecting daily life.

Rubbers, in general, were not known as industrial materials until vulcanization processes were discovered, which opened up economic prospects for the use of rubber manufactured goods [1].

Due to their market demand, tires are considered to be one of the main products manufactured from rubber. Generally, tires are prepared using several layers of rubber and many other compounds, especially carbon black, which help to form a composite structure with suitable thickness [2].

The quality of tires depends mainly on the ratio of the different compounds used during their production and the vulcanization process details, such as the type of materials used, the time of the applied process and the temperature used; this improves the tensile strength, elasticity and other properties of the rubber [3,4].

Rubbers have unique properties and therefore, they have no alternatives. On the other hand, almost 70% of rubber is used in the manufacturing process of the tires industry [5]; despite this, millions of tons of rubber tire scrap are generated. Tire scrap is completely resistant to all types of degradation due to the presence of different additives and the three-dimensional cross-linked chains formed after the vulcanization process [6,7].

Furthermore, it is important to reclaim rubber from tire scrap to reduce environmental problems and to preserve rubber raw materials. The main problems associated with reclaimed rubber are the treatment processes of tire scrap. Some processes depend on combustion or pyrolysis of tires, either for the production of oils as liquid hydrocarbons or the production of energy [8] and, in both processes, rubber is lost and is not compensated for during the processes. Scrap tire is sometimes used without treatment, however, the

granulate materials that release pollutants and require curing are a problem [9]. Other methods, such as mechanical [10], thermal−mechanical [11] or mechano-chemical [12] methods concentrate on the scrap tire without any treatment. Furthermore, rubber raw material is lost when using these methods, and at the same time, some ecological problems are caused because the rubber releases pollutants into the environment; thus, scrap tire needs to be treated [13].

Accordingly, in the following work, a chemical dissolution process was used to recover rubber and carbon black from scrap tire. It is important to highlight that in the applied treatment process, hydrocarbon was used as the solvent in the rubber's recovery process, and gas oil, commonly known as diesel fuel, was selected due to its ability to penetrate between the vulcanized rubber chains. In addition, 4-hydroxy-TEMPO was used as the catalyst to help the rubber chains dissolve faster in the gas oil and to produce a homogeneous solution in a short time-period at an ambient temperature.

## 2. Materials and Methods

### 2.1. Materials

Scrap truck tires from a local market were collected and prepared for treatment by first, cutting them into pieces, cleaning them and then grinding them into particles sized between 135 and 141 nm. Gas oil was obtained from a gas station, which was supplied by the Baiji oil refinery. 4-Hydroxy-2,2,6,6-tetramethyl pipridine 1-oxyl (4-Hydroxy-TEMPO) was obtained from Sigma−Aldrich (Burlington, MA, USA). Aluminum oxide and methanol were supplied by the BDH Chemical Company (Poole, UK) and were used as received. Montmorillonite clay was collected from the north of Mosul.

### 2.2. Chemical Dissolution Process

A dissolution process was used to reclaim the two main composite materials of the tires (i.e., the rubber and carbon black). First, 100 mL of gas oil acting as the hydrocarbon solvent was poured into a 250 mL conical flask with a stopper, and then 5.0 g (5.0 wt%) of scrap tire powder was added. The flask was closed and placed inside a water bath fixed at 50 °C. The conical flask was kept for three days and shaken up from time to time inside the water bath.

#### 2.2.1. Reflux of the Soaked Solution

The soaked solution was poured into a 250 mL round bottom flask, in which 0.5 g of 4-Hydroxy-TEMPO acting as a catalyst was added along with 1.0 g of alumina acting as the catalyst support. The flask contents were heated to the boiling point of gas oil (in the range of 250–350 °C). After two hours of heating the contents, the process was stopped, left to cool down, and then filtered to get a clear black solution.

#### 2.2.2. Montmorillonite Clay Separation Method

The suspended carbon black in the gas oil solution did not precipitate; therefore, a large column was filled with montmorillonite powder and used for purification. The clear black solution was allowed to pass through the montmorillonite powder slowly and in a controlled manner. The output was a transparent solution of rubber dissolved in gas oil solvent. The black montmorillonite clay was washed several times with clean gas oil solvent, which quickly passed through the column, taking carbon black with it, and the pure montmorillonite clay could be used again. At the same time, the collected black gas oil solvent was left to precipitate the carbon black for one day, and was then filtered and dried for the next treatment.

### 2.2.3. Rubber Precipitation

The precipitator methanol and (25–30%) *v/v* was added to the rubber/gas oil solution, and then the final solution was transferred into an open pan and exposed to sunlight in an open space in order to avoid direct heat (vacuum oven at 40 °C and 0.01 bar can be used). A layer of pure rubber precipitated on the pan after some time. A brownish-white precipitate was collected and prepared for characterization.

### 2.2.4. Treatment of the Carbon Black

The carbon black montmorillonite clay was washed with pure gas oil, and the collected black gas oil solvent was left for one day for CB precipitation. The CB was filtered, dried and pyrolyzed using a tube furnace, which was fixed at 500 °C ± 10 °C, for the removal of the remaining rubbers and other additives. The sample was heated inside the furnace for one hour, and the produced carbon black was left to cool and kept inside a closed sample tube for analysis.

### *2.3. Characterization of Pristine and Recovered Rubber and Carbon Black*
### 2.3.1. FTIR Spectroscopy

For the FTIR spectrum of the scrap tire sample before treatment, the rubber and carbon black were recovered using an FTIR spectrophotometer model JASCO V-630, Portland, OR 97211, United State, in the spectral range of 400–4000 cm$^{-1}$.

### 2.3.2. X-ray Diffraction (XRD)

The X-ray diffraction of the pristine sample and the recovered rubber and carbon black was measured using a Philips X-ray diffractometer (PW 1730), Amsterdam, the Netherlands, with a Cu-Ka radiation target and nickel filter, at a current of 30 Kv and under a voltage of 30 mA.

### 2.3.3. Thermal Analysis

The TGA, DTA and DSC characterizations of the different samples were measured using a TA Instruments DSC SDT Q600 (University of Buffalo, Buffalo, NY 14260, USA), at a heating rate of 80 °C/min and within a temperature range of 25–1000 °C, using an $Al_2O_3$ reference sample.

### 2.3.4. Scanning Electron Microscope (SEM)

The SEM images of different samples were recovered using a TESCAN MIRA FESEM (Brno, Czech Republic). The measured samples were mounted on studs using double face adhesive tape and exposed to a gold ion beam sputter under vacuum.

### 2.3.5. BET Measurements

The specific surface area, Langmuir surface area (i.e., lang), pore area and the multilayer formation in the micropores of pyrolyzed carbon black were measured using a BELSORP MINI 11, surface area and porosimetry analyzer (Osaka, Japan).

### 2.3.6. $^1$H NMR Measurement

The $^1$H NMR of the scrap tire before treatment and the recovered rubber was done using a spectrometer (Bruker BioSpin GmbH, Ettlingen, Germany) at 400 MHz frequency, with deuterated DMSO used as the solvent.

## 3. Results and Discussion

The recovery of rubber from the scrap tire and the recycling of carbon black are important and urgent concerns due to environmental pollution issues and the need to preserve rubber raw materials.

In this study, a chemical dissolution process was used for the recovery of rubber using gas oil as the solvent; this was performed in the presence of a catalyst, which is an important issue to discuss. At the same time, the obtained results show that the applied processes are simple, economic and safe, and the recovered materials retain their qualities and properties.

### 3.1. Characterization of Scrap Tire Powder

Scrap tire in its powder form was analyzed in order to determine the characteristics of the recovered materials.

### 3.1.1. FTIR Study

The FTIR absorption frequencies of the featured functional groups of pristine samples were studied.

The FTIR spectrum (Figure 1) shows that the characteristic functional groups of the original scrap tire sample mainly consist of hydrocarbon groups that appear at 3082 cm$^{-1}$ and 1450 cm$^{-1}$, and these represent $v$(C-H)$_{str}$ and $v$(C=C)$_{str}$ in the aromatic ring of styrene [8].

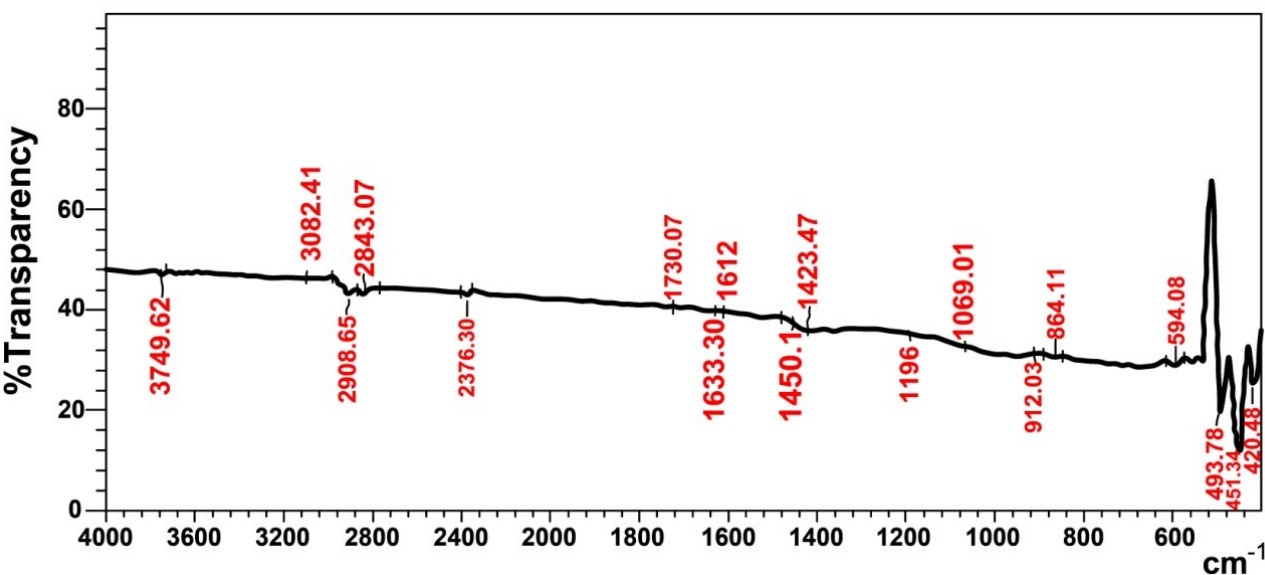

**Figure 1.** FTIR spectra of original scrap tire rubber before treatment.

The absorption frequencies at (2920 and 2843) cm$^{-1}$, in addition to those at 912 cm$^{-1}$ and 522 cm$^{-1}$, represent the asymmetric and symmetric bands of $v$(C-H)$_{str}$, $v$(C-H)$_{wag}$ and $v$(C-H)$_{twist}$, respectively. The absorption bands at 1730 cm$^{-1}$ and 1633 cm$^{-1}$ overlap with those at 1612 cm$^{-1}$, and represent $v$(C=O)$_{str}$ and $v$(C=C)$_{str}$ alkene of carbon black. The absorption frequencies at 1196 cm$^{-1}$ and 1069 cm$^{-1}$ represent $\delta$(C-H) aliphatic and $\delta$(C-H) in the plane [14]. The absorption bands at 594 cm$^{-1}$ and (451 and 494) cm$^{-1}$ belong to $v$(C-S)$_{str}$ and $v$ (S-S)$_{str}$, which represent sulfide and disulfide groups in the vulcanized rubber, respectively [15]. The aforementioned absorption frequencies indicate that the scrap tire mainly consists of styrene−butadiene rubber, natural rubber and carbon black.

### 3.1.2. $^{1}$H NMR Spectra Study

The $^{1}$H NMR spectrum of the scrap tire sample before treatment (Figure 2) shows resonance at 0.87–2.01 ppm, which represents the paraffinic protons of butadiene and natural rubber. The resonance at 7.17 ppm represents the aromatic protons of styrene [8]. The NMR peaks show the scrap tire sample, which mainly consists of natural and SBR rubber.

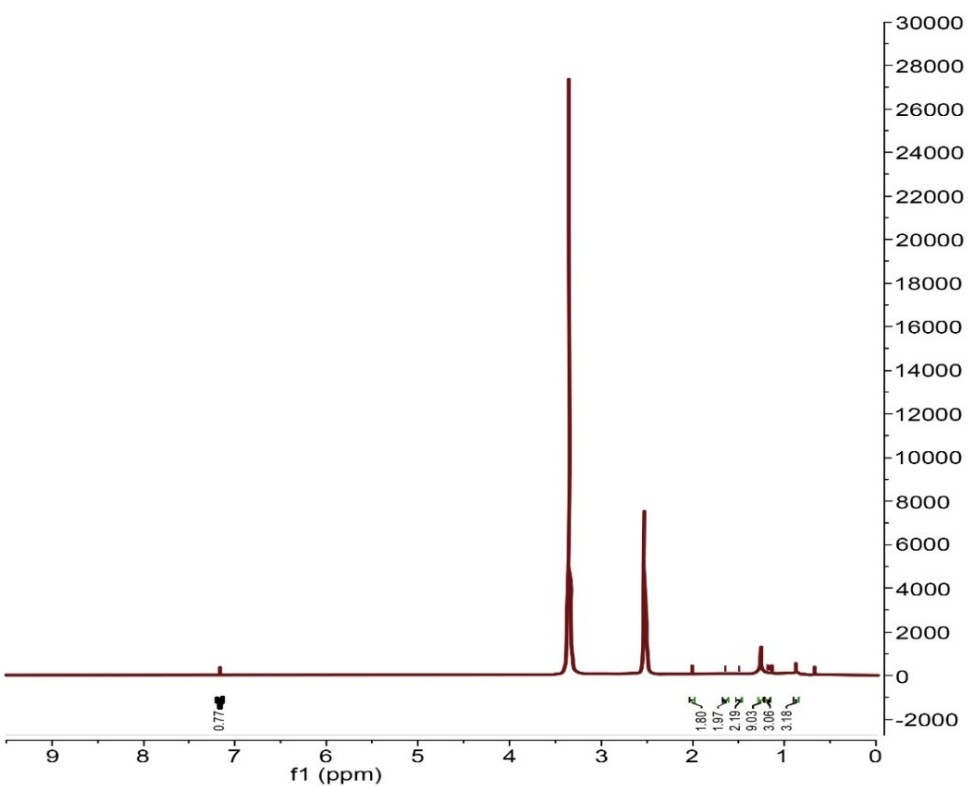

**Figure 2.** [1] H NMR spectra of original scrap tire rubber before treatment.

### 3.1.3. XRD Analysis

An X-ray diffraction analysis of the pristine tire sample was performed. The area under the intensity curve can be used to determine the crystallinity percentage (%$Xc$), and this was calculated according to the following equation [16]:

$$\% \ Xc = \frac{Ac}{Aa + Ac} \ast 100 \tag{1}$$

where Ac represents the crystalline phase and Aa represents the amorphous phase.

The XRD pattern (Figure 3) and the XRD data (Table 1) show many XRD peaks: most are intensive peaks, and one broad band is back to styrene rubber, which has a sharp peak that appears at 19.8467° along the 2theta axis, with a crystallinity percentage of 92.89%. The carbon black appears with intensive peaks at 22.8°, 26.5°, 29.3° and 44.5° along the 2theta axis, with crystallinity percentages of 79.7%, 100%, 82.7% and 24.0%, respectively.

### 3.1.4. Thermal Analysis

The TGA, DTA and DSC thermograms of the original scrap tire sample were studied; these are shown in Figure 4 and Table 2. The weight loss percentage in the TGA thermogram (Figure 4a) of the initial and final decomposition temperature is 3.0% at 145 °C and 54.5% at 520 °C, indicating that the pristine tire is thermally stable, whereas the maximum decomposition temperature Tmax shows a 37.5% weight loss at 382 °C, as shown in Table 2 and Figure 4a.

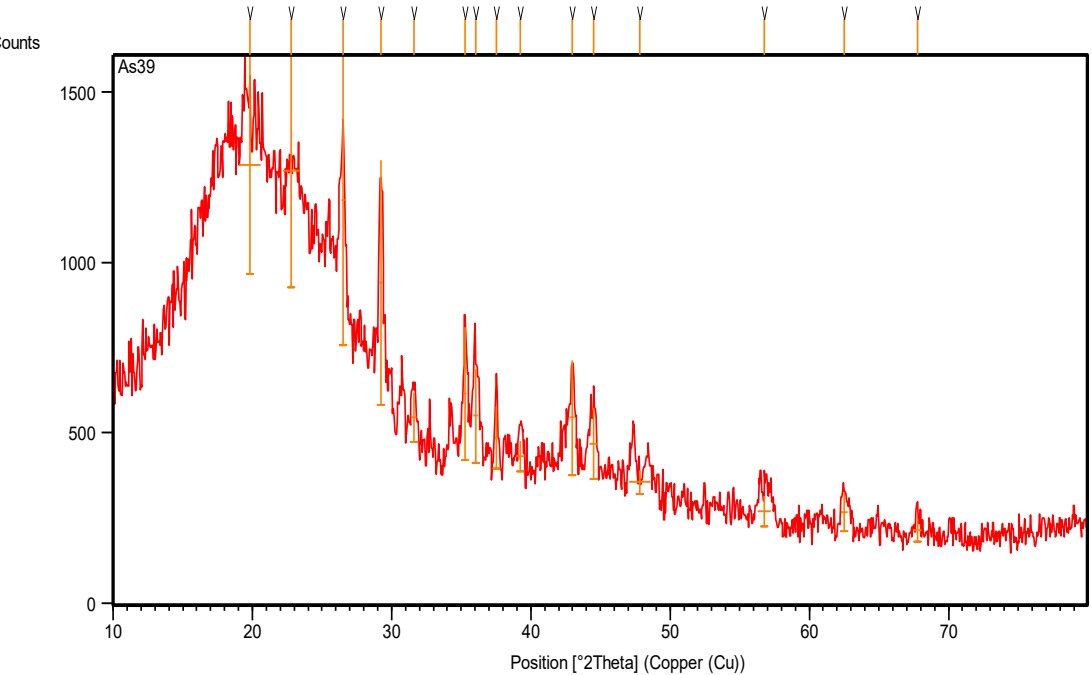

**Figure 3.** XRD pattern of original rubber scrap tire before treatment.

**Table 1.** XRD intensity scan data of original scrap tire before treatment.

| Peak Position at 2θ° | Crystallinity Percentage % Xc |
|---|---|
| 19.8467 | 92.89 |
| 22.7596 | 79.72 |
| 26.5107 | 100.00 |
| 29.2505 | 82.67 |
| 31.5896 | 16.90 |
| 35.2853 | 44.76 |
| 36.0450 | 31.52 |
| 39.2250 | 10.14 |
| 37.5078 | 20.74 |
| 42.9765 | 38.69 |
| 44.4866 | 23.99 |
| 47.7969 | 7.82 |
| 56.7928 | 10.43 |
| 62.5036 | 13.08 |
| 67.7690 | 7.68 |

**Table 2.** TGA, DTA and DSC thermal analyses of pristine tire rubber and pyrolyzed carbon black recovered from scrap tire.

| Sample | TGA Weight (%) | | | | DTA Decomposition Rate (°C. min/mg) | | | DSC (W/g) | | | |
|---|---|---|---|---|---|---|---|---|---|---|---|
| | IDT | FDT | Tmax | Tcr | | | | Tg °C | ΔHf (J/g) | | |
| Pristine Tire Rubber | 3.0 145 °C | 54.5 520 °C | 37.5 382 °C | 52.0 480 °C | 0.26 68 °C | 0.12 131 °C | 0.98 353 °C | <0 | −1775 76.7 °C | +550.6 395 °C | −2075 693.5 °C |
| Pyrolyzed Carbon Black | 0.1 40 °C | 38.8 1000 °C | 18.5 710 °C | 22.3 755 °C | 0.143 80.9 °C | 0.08 144 °C | 0.30 255 °C | / | −255 80.3 °C | −2173 687.0 °C | / |

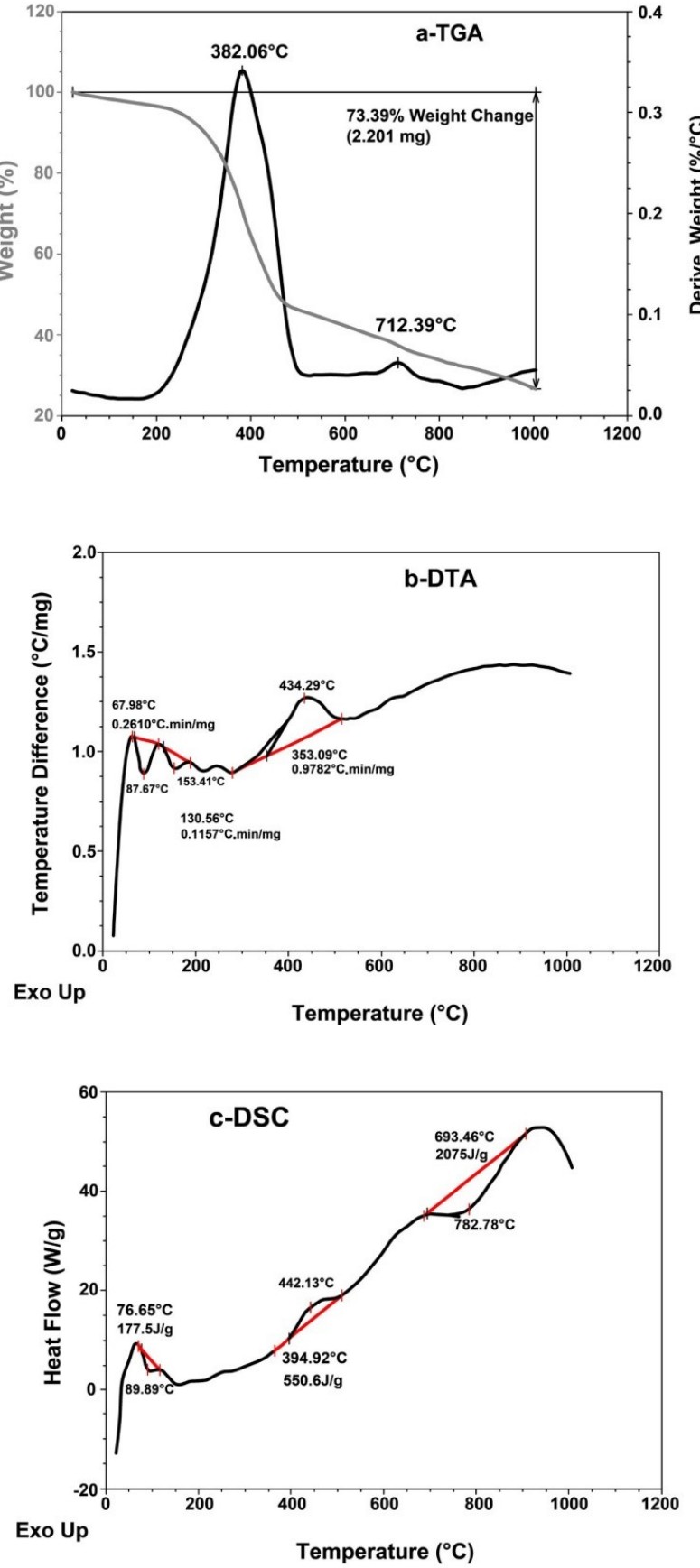

**Figure 4.** (**a**) TGA, (**b**) DTA and (**c**) DSC thermal analysis thermograms of original scrap tire sample.

The crystalline decomposition temperature ($T_{cr}$) shows a 52.0% weight loss at 480 °C, which means that the rubber in the pristine tire has lost half of its weight at $T_{max}$ (382 °C), and that the crystalline structure of the tire composite collapses at $T_{cr}$ = (480 °C); this indicates that the pristine tire is thermally stable.

The DTA thermogram (Figure 4b and Table 2) shows three maxima at 68 °C, 131 °C and 353 °C, with rubber decomposition rates of 0.26, 0.12 and 0.98 °C.min/mg, respectively, which means that the rate of rubber decomposition increases with an increase in temperature. However, the rate of rubber decomposition is still low at a high temperature. The first two maxima represent free and bond water, whereas the third maximum represents rubber decomposition.

The DSC thermogram (Figure 4c and Table 2) shows that the glass transition temperature $T_g$ is below zero [17]. The pristine tire shows three enthalpy of fusion (heat of fusion) ($\Delta H_f$) magnitudes at 76.7 °C, 395 °C and 693 °C. The first two magnitudes (i.e., −177.5 J/g and +550.6 J/g) are exothermic and endothermic, respectively, and they represent free and bond water in rubber of tires, while the third (i.e., −2075 J/g) is highly exothermic and represents the crystalline composite of the carbon black in pristine tire.

### 3.1.5. SEM and BET Studies

The SEM images of the original scrap tire with different magnifications (Figure 5) show that the sample is a rubbery composite, which consists of homogeneous components with non-uniform surfaces filled with holes and folds. The surface morphologies of the sample (Figure 5a,b) show white portions with luminous edges, representing crystalline particles mixed homogeneously with rubber due to the carbon black and other inorganic additives present in the sample, which give the sample molar cohesion and elastic properties [18].

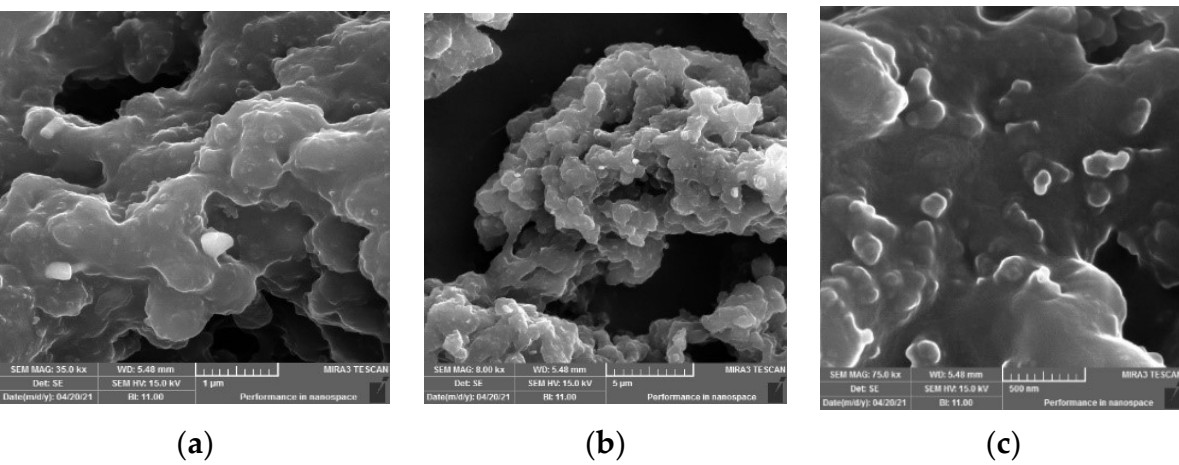

    (**a**)                        (**b**)                        (**c**)

**Figure 5.** SEM images with different magnifications (**a**) 35k, (**b**) 8k, (**c**) 75k of original scrap tire sample.

The SEM image (Figure 5c) shows that the sample is in an elastic form with small particles distributed in and between the folds in clusters and is a carbon black nano composite.

The Brunauer−Emmett−Teller (BET) measurements of the original scrap tire (Table 3) show the specific surface area of the particles. The Langmuir plot describes the total amount of nitrogen gas bond to the particles' surfaces at a specific temperature and at a pressure of 1.7959 m$^2$/g. The t plot provides information on the multilayer formation of the micropores; the micropore surface area (0.2705 m$^2$/g) was calculated from the difference between the BET area and the external surface area.

Finally, the BJH plot (Table 3) yields the pore area of 3.09 nm and the specific surface area of 1.2749 cm$^2$/g.

From the BET plot (Table 3), the specific BET area is 1.0249 cm$^2$/g, the total volume is 0.0072 cm$^3$/g and the mean pore diameter is 28.228 nm. The previous BET data show that the tire rubber is present in a large area of folded surface that includes nanopores interspersed with nano-sized carbon black particles.

**Table 3.** BET shows the specific surface area of original scrap tire particles.

| BET Plot | | |
|---|---|---|
| $V_m$ | 0.2355 | [cm$^3$ (STP) g$^{-1}$] |
| $a_{s,BET}$ | 1.0249 | [m$^2$ g$^{-1}$] |
| Total pore volume ($p/p_0$ = 0.990) | 0.0072325 | [cm$^3$ g$^{-1}$] |
| Mean pore diameter | 28.228 | [nm] |
| **Langmuir Plot** | | |
| $V_m$ | 0.4126 | [cm$^3$ (STP) g$^{-1}$] |
| $a_{s,Lang}$ | 1.7959 | [m$^2$ g$^{-1}$] |
| **t Plot** | | |
| **Plot Data** | **Adsorption Branch** | |
| $a_1$ | 0.2705 [m$^2$ g$^{-1}$] | |
| $V_1$ | 0 [cm$^3$ g$^{-1}$] | |
| **BJH Plot** | | |
| **Plot Data** | **Adsorption Branch** | |
| $V_p$ | 0.0072414 [cm$^3$ g$^{-1}$] | |
| $r_{p,peak}$ (*Area*) | 3.09 [nm] | |
| $a_p$ | 1.2749 [m$^2$ g$^{-1}$] | |

### *3.2. Recovery of Rubber*

Rubber and carbon black are the main components of tires and in this study, they were recovered from scrap tire through a chemical dissolution process, using gas oil as the hydrocarbon solvent, in the presence of 4-Hydroxy-TEMPO as a catalyst.

The rubber was recovered from the scrap tire after being soaked for 72 h in gas oil solvent. The sample was kept under shaking conditions in bath water at 50 °C.

A weight percentage of 5 wt% of scrap tire powder in gas oil was found to be a suitable dilution of the tire sample in gas oil, helping the solvent molecules more easily penetrate between the rubber chains [19]. The 4-Hydroxy-TEMPO catalyst was found to be more active in the recovery of the rubber and CB from the tires when compared with the non-catalytic process. This was demonstrated by the fact that the percentage yield of recovered rubber from a non-catalytic process is 0.86 g (represents 34.4% of its real percentage in the tire), compared to 0.955g (represents 38.2% of its real percentage in the tire) from the catalytic process. The recovered rubber was characterized using the following analysis.

### 3.2.1. FTIR Study

The characteristic functional groups of the recovered rubber (Figure 6) show absorption frequencies at 3065 cm$^{-1}$ (756 and 902 cm$^{-1}$) and 1454 cm$^{-1}$, which represent the $v$(C-H)$_{str}$ and $v$(C=C)$_{str}$ aromatic rings of styrene, respectively in SBR rubber. The band that appears at 1709cm$^{-1}$ belongs to $v$(C=C)$_{str}$ of the unsaturated olefin of natural and butadiene rubber [20]. The aliphatic bands at 1373 cm$^{-1}$, (2920 and 2855) cm$^{-1}$, 1183 cm$^{-1}$ and 1018 cm$^{-1}$ represent $v$(CH$_3$)$_{str}$, $v$(C-H)$_{str}$, $\delta$(C-H) aliphatic and $\delta$(C-H) in the plane, respectively, and they all belong to styrene−butadiene and natural rubber. Finally, the absorption bands at 598 cm$^{-1}$ and (444 and 501) cm$^{-1}$, belong to $v$(C-S)$_{str}$ and $v$ (S-S)$_{str}$,

which represent the sulfide and disulfide groups in the rubber sample, respectively; this means that a trace of carbon black still remains in the recovered rubber.

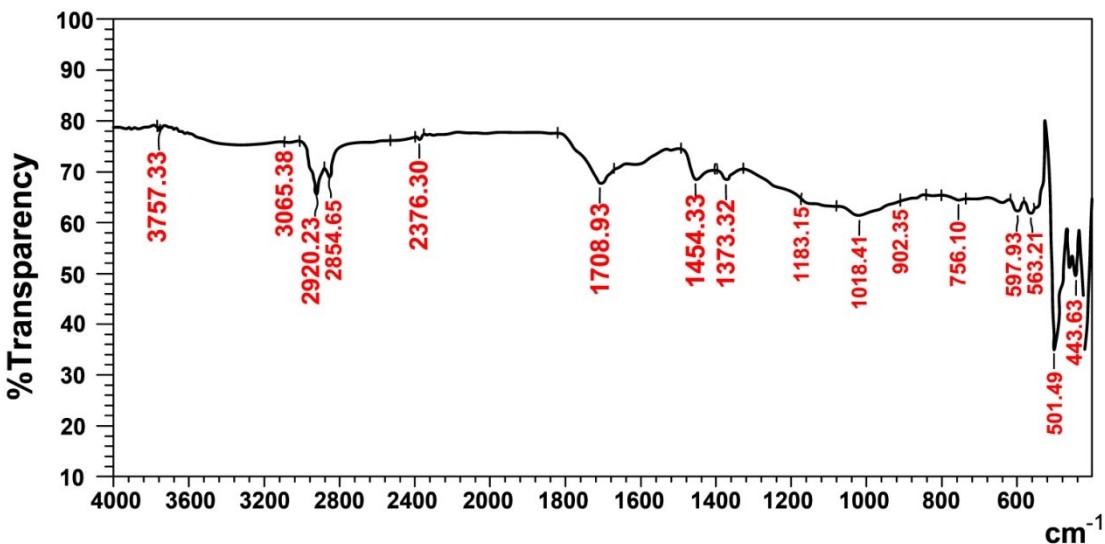

**Figure 6.** FTIR spectra of recovered rubber.

### 3.2.2. $^1$H NMR Study

The $^1$H NMR spectrum of the recovered rubber (Figure 7) shows resonance (3H, m) at 3.31 and 4.94 ppm, which represents the unsaturated ethylene groups of butadiene and isoprene rubber; whereas, the resonance (9H, m) observed at 0.78–2.11 ppm represents the methyl, methylene and methine protons of isoprene, styrene and butadiene rubber. In addition, the resonance (3 H, m) observed at 2.25–2.44 ppm belongs to the methyl group of isoprene. Finally, the resonance observed at 7.25 ppm (Figure 7) represents the aromatic protons of the styrene ring.

### 3.2.3. XRD Study

The X-ray diffraction pattern of the recovered rubber (Figure 8) shows many intense peaks, which is different from those observed in the original tire sample in Figure 3, where their position is along the 2θ axis.

The XRD intense data, listed in Table 4, show that the recovered rubber has a lower number of crystalline peaks, which means that its purity is increased, and the crystalline peaks of styrene have become clearer.

The crystallinity percentage of the intense peaks of the recovered rubber (Figure 8 and Table 4), which were calculated according to Equation (1), increased; the XRD peaks showed high altitude with greater full-width at half-maximum (FWHM), and even the d-spacing (the interatomic spacing) increased after recovering the rubber. The broad band of styrene with an intense peak is positioned at 18.6° with 100% crystallinity. This band is highly focused, whereas the peaks at 38.5° show high crystalline percentages and represent the crystalline structure of the rubbers after purification. Nevertheless, some peaks of carbon black were still present in the recovered rubber.

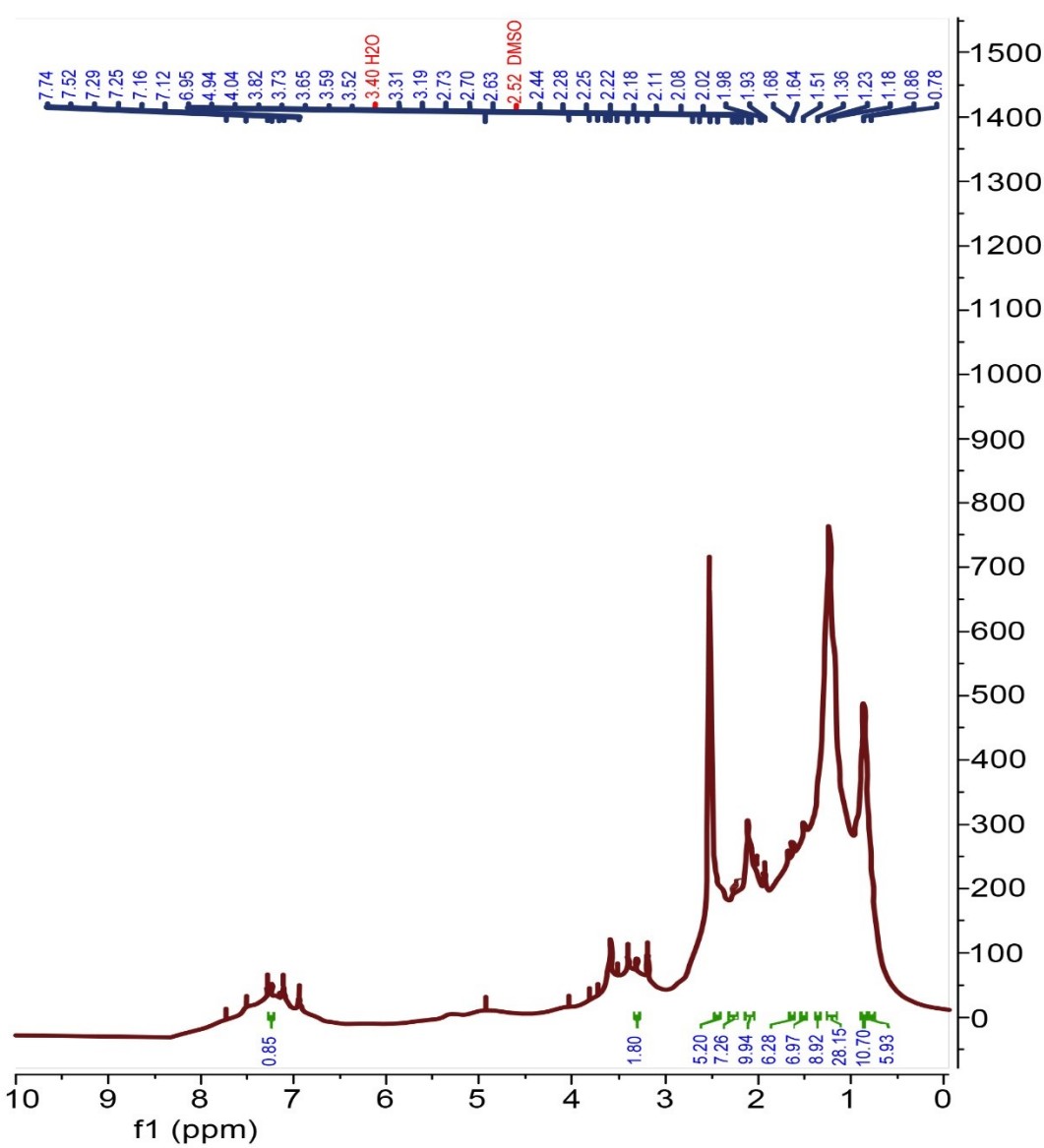

**Figure 7.** $^1$H NMR spectrum of recovered rubber.

**Table 4.** XRD intensity scan data of recovered rubber.

| Peak Position at 2θ° | Crystallinity Percentage % Xc |
|---|---|
| 18.5718 | 100.00 |
| 26.6185 | 72.88 |
| 29.4586 | 81.23 |
| 35.7961 | 18.72 |
| 38.5113 | 97.39 |
| 44.7209 | 48.72 |
| 47.9434 | 11.61 |
| 65.1406 | 24.18 |
| 78.1903 | 26.17 |

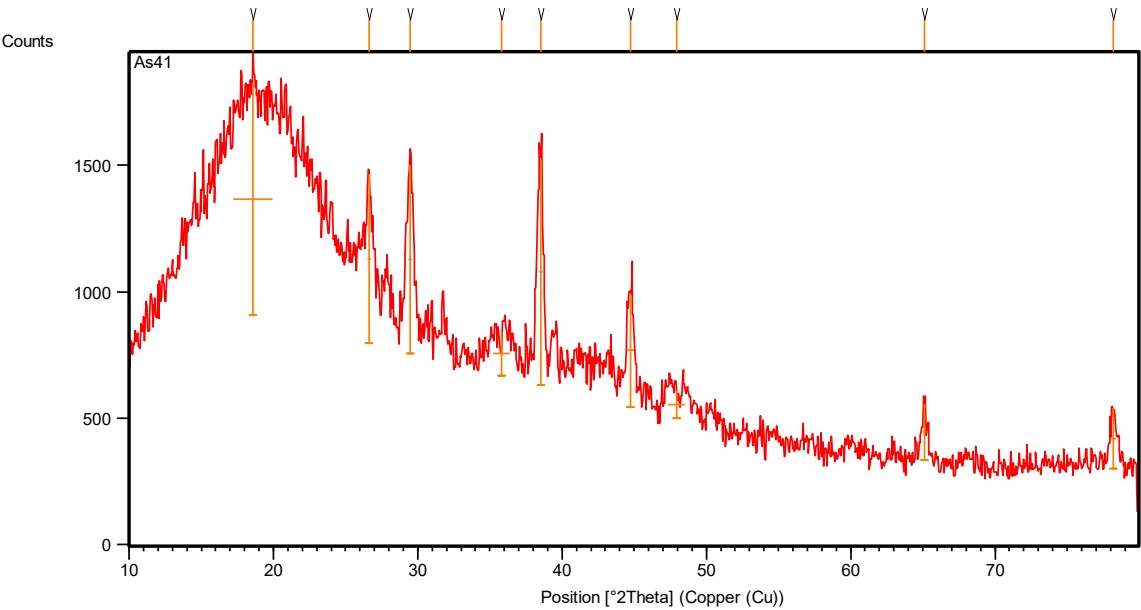

**Figure 8.** XRD pattern of recovered rubber.

### 3.2.4. SEM Study

The SEM images of the recovered rubber sample (Figure 9) show that the sample has lost most of its crystalline particles derived from additives. The SEM image (Figure 9a) shows the surface morphology of the sample, which appears as a rubbery aggregation with disconnected pieces; this is due to the loss of its vulcanization and most of its filling and additive materials after being dissolved in gas oil. The images (Figure 9b,c) show very pure particles, which have corrugated surfaces with protrusions.

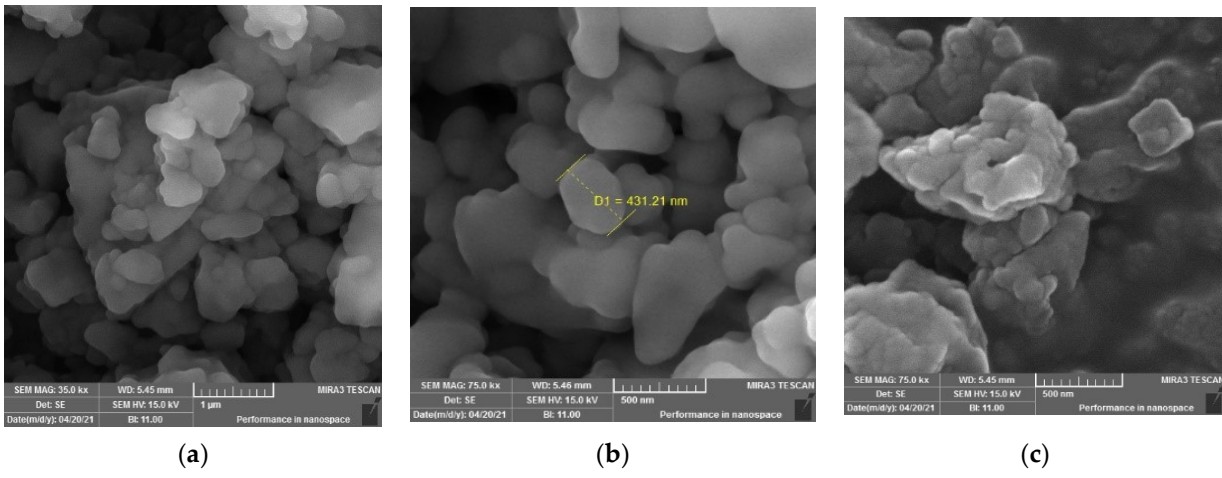

**Figure 9.** SEM images of recovered rubber with different magnifications (**a**) 35k, (**b**) 75 k, (**c**) 75 k.

### 3.3. Recovery of Carbon Black

The carbon black (CB) was collected mainly after filtration of the recovered rubber solution using montmorillonite clay. It was then thermally treated using the pyrolysis method at 500 °C inside a tube furnace for one hour. The 5.0 g of tire waste yielded 0.3 g of pure CB (representing approximately 17.0% of its real percentage in the tire). Characterizations of pyrolyzed CB were performed to determine the crystalline structure, thermal behavior and surface morphology of particles with specific surface areas.

### 3.3.1. XRD Study

The X-ray diffraction pattern of pyrolytic CB (Figure 10, Table 5) shows many crystalline peaks with high intensity, located at 25.36°, 26.44°, 29.34°, 35.27° and 43.02° along the 2θ axis, representing the main crystalline peaks of CB after pyrolysis [21]. The main observations include dwarfism of the crystalline rubber peak at 38.5° and its crystalline percentage of only 4.5%.

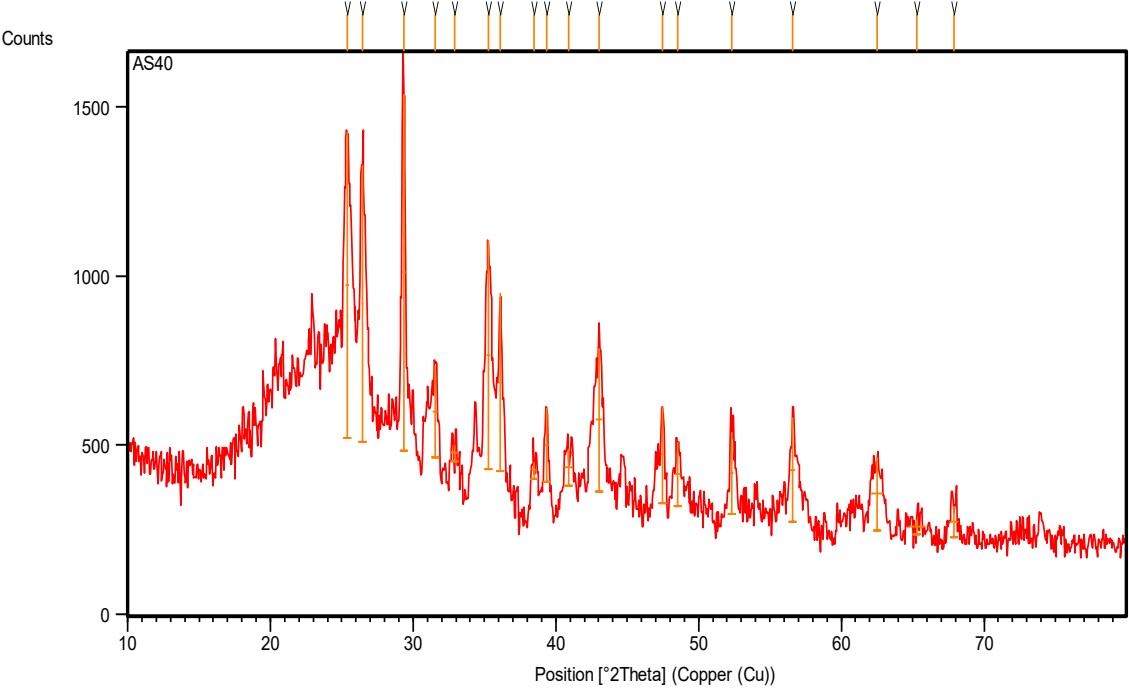

**Figure 10.** XRD pattern of pyrolytic carbon black recovered from scrap tire.

**Table 5.** XRD intensity scan data of pyrolytic carbon black recovered from tire waste.

| Peak Position at 2θ° | Crystallinity Percentage % Xc |
|---|---|
| 25.3675 | 85.82 |
| 26.4415 | 77.54 |
| 29.3373 | 100.00 |
| 31.5279 | 25.54 |
| 32.9148 | 4.70 |
| 35.2753 | 64.12 |
| 36.1235 | 50.07 |
| 38.4956 | 4.54 |
| 39.3355 | 20.67 |
| 40.9126 | 10.52 |
| 43.0155 | 40.03 |
| 47.4737 | 27.22 |
| 48.5396 | 17.97 |
| 52.3163 | 23.07 |
| 56.6060 | 29.06 |
| 62.4803 | 20.52 |
| 65.2697 | 3.96 |
| 67.8833 | 8.90 |

The pyrolysis of CB at a high temperature removed all the remaining hydrocarbon materials and other inorganic additives in the sample. Figure 10 and the data recorded in Table 5 show a high number of crystalline peaks for CB after pyrolysis due to the different d-spacing (lattice spacing) in its crystals, which represents the distance between the planes of carbon atoms that form many diffraction peaks [22].

### 3.3.2. Thermal Study

The TGA, DTA and DSC thermograms of pyrolytic CB (Figure 11) were studied. The TGA thermogram (Figure 11a and Table 2) at IDT shows a very low weight loss % of 0.1% starting at 40 °C; however, at FDT, although the temperature reaches 1000 °C, the weight loss percentage is still low and does not reach more than 38.75%, which means the CB sample is thermally stable. The $T_{max}$ of the CB sample, calculated from the TGA thermogram (Figure 11a), and records in Table 2 show a low weight loss percentage of 18.5% at 710 °C. In addition, the $T_{cr}$ represents the temperature at which the crystalline structure of CB collapses. The sample loss is 22.3% of its weight at 755 °C, which also means that the recovered and pyrolytic CB has a stable thermal structure.

The DTA thermogram (Figure 11b and Table 2) gives three maxima at 80.9 °C, 144 °C and 255 °C, with very low decomposition rates of 0.143, 0.08 and 0.30 °C.min/mg, respectively; this shows that the sample is stable and lost its free and bound water in the beginning, and then it starts, at a low rate, with its composite structure. The heat of fusion $\Delta H_f$ in the DSC thermogram (Figure 11c and Table 2) is 255 J/g at 80.3 °C, which shows that the free and bound water are a part of the sample composite. The second $\Delta H_f$ of $-2173$ J/g of high value refers to the collapse of the CB crystalline structure at 687 °C.

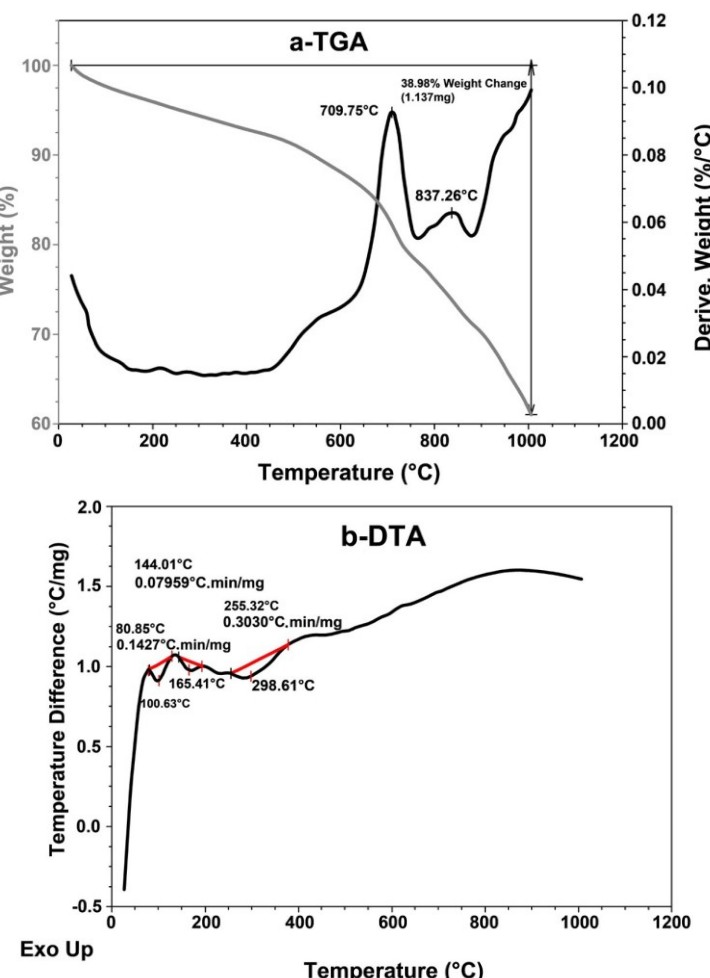

**Figure 11.** *Cont.*

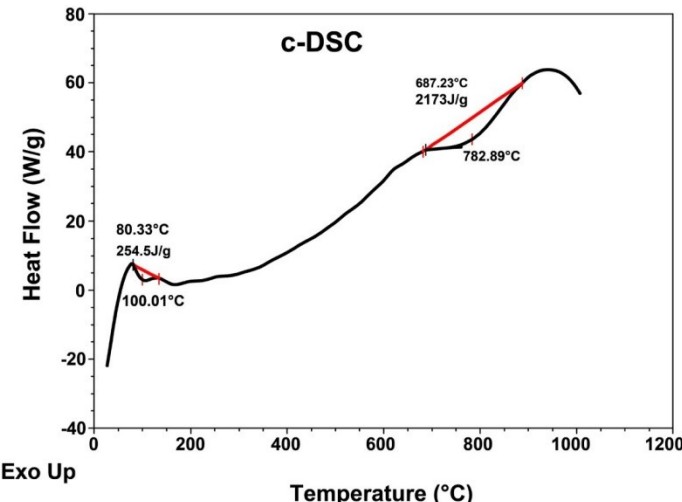

**Figure 11.** (**a**) TGA, (**b**) DTA and (**c**) DSC thermal analysis thermograms of pyrolytic carbon black recovered from scrap tire.

### 3.3.3. SEM and BET Study

The SEM images of the pyrolytic carbon black (Figure 12) show the morphological surfaces of the CB particles. In Figure 12a, the image shows that the CB particles in the nanoscale composite gather in clusters and have spherical shapes, smooth surfaces and pure particles with no rubber or other additives. Even the image with greater magnification (Figure 12b), shows that the CB particles are present as a nanocomposite with sharp edges and a crystalline structure. The SEM images show that the pyrolytic CB particles are spherical in shape and nano-sized, and the whitening surface of the particles indicates the presence of crystallite parts on their morphological surfaces.

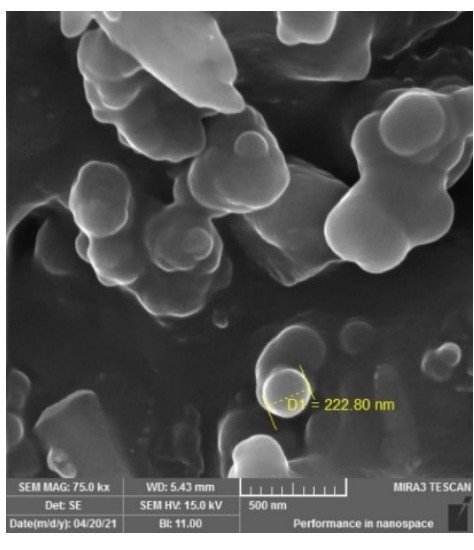

(**a**)

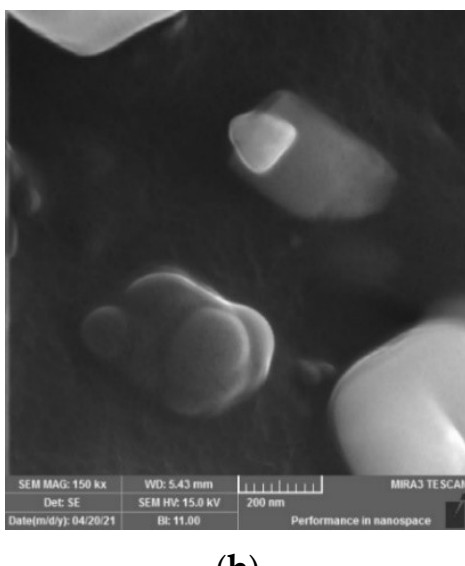

(**b**)

**Figure 12.** SEM images of pyrolytic carbon black with different magnifications (**a**) 75k, (**b**) 150k.

The BET measurement (Table 6) of pyrolytic CB shows the Langmuir surface area is 46.671 m$^2$/g, which is high when compared with the surface area of pristine tire particles. Even the t plot (Table 6) shows a high surface area (i.e., 36.414 m$^2$/g), which means that the CB particles can achieve high physical adsorption. In addition, the pore area from BJH (Table 6) is observed as very narrow (1.85 nm) with a small volume (0.267 cm$^3$/g). Finally, Table 6 shows the specific BET area of 44.853 m$^2$/g, the total pore volume of 0.268 cm$^3$/g

and the mean pore diameter of 23.9 nm. A pore diameter less than 50 nm, according to the IUPAC nomenclature [23], are a nanoporous material. The CB with 23.9 nm is a mesoporous material, and it has porosity within the mesopore range, which increases its specific surface area. All the aforementioned parameters show that pyrolytic CB particles have high specific surface areas with pores that improve gas or liquid adsorption capacity, which represents the standard specifications for industrial carbon black.

**Table 6.** BET data shows the specific area of pyrolytic carbon black.

| **BET Plot** | | |
|---|---|---|
| $V_m$ | 10.305 | [$cm^3$ (STP) $g^{-1}$] |
| $a_{s,BET}$ | 44.853 | [$m^2 \, g^{-1}$] |
| Total pore volume ($p/p_0 = 0.990$) | 0.2681 | [$cm^3 \, g^{-1}$] |
| Mean pore diameter | 23.914 | [nm] |
| **Langmuir Plot** | | |
| $V_m$ | 10.723 | [$cm^3$ (STP) $g^{-1}$] |
| $a_{s,Lang}$ | 46.671 | [$m^2 \, g^{-1}$] |
| **t Plot** | | |
| **Plot Data** | **Adsorption Branch** | |
| $a_1$ | 36.414 | [$m^2 \, g^{-1}$] |
| $V_1$ | 0 | [$cm^3 \, g^{-1}$] |
| **BJH Plot** | | |
| **Plot Data** | **Adsorption Branch** | |
| $V_p$ | 0.2686 | [$cm^3 \, g^{-1}$] |
| $r_{p,peak}$ (*Area*) | 1.85 | [nm] |
| $a_p$ | 51.556 | [$m^2 \, g^{-1}$] |

## 4. Conclusions

The recovery of rubber and CB from scrap tire was performed through a high dissolution process, with gas oil solvent used in the presence of a 4-Hydroxy-TEMPO solution. The gas oil solvent showed high activity in the dissolving of the rubber chains, but left the CB particles and other inorganic additives insoluble. The activity of the gas oil solvent was elevated in the presence of the 4-Hydroxy-TEMPO catalyst, where the latter increased the homogeneity of the rubber in the gas oil solvent, and played a positive role as a rubber polymerization inhibitor at a high temperature [24,25]. Montmorillonite clay showed high efficiency in the separation process of the rubber−gas oil solution, where the filler material from the CB and other additives was used in tire manufacturing. 5.0 g of the scrap tire sample in 100 mL of gas oil reclaimed 0.955 g of pure rubber and 0.3 g of pure CB, which represents 38.2%w/w of the real percentage of rubber in the tire's waste and 17.0%w/w of the real percentage of CB in the tire's waste, respectively. The specifications of the recovered rubber and CB by chemical dissolution process have shown same to that of the native rubber and CB used in the manufacturing of tires according to the chemical, physical and morphological analyses, which have been conducted extensively. The gas oil solvent and montmorillonite clay were recycled and reused after purifications.

**Author Contributions:** Collecting and arranging the literature, A.S.A. Research idea-generating, F.H.J. Conducting the experimental work, A.S.A. Analyzing, discussing and interpreting the idea, F.H.J. and A.S.A. Checking the final review of the draft, F.H.J. All authors have read and agreed to the published version of the manuscript.

**Funding:** This research received no external funding.

**Institutional Review Board Statement:** Not applicable.

**Informed Consent Statement:** Not applicable.

**Data Availability Statement:** The data presented in this study are available on request from the corresponding author.

**Acknowledgments:** The authors would like to thank Thomas Seidensticker from the Lab. of Industrial Chemistry, TU Dortmund University, Germany, for providing the catalyst. The authors are also thankful to the University of Mosul, Iraq, for providing the facilities to carry out this work.

**Conflicts of Interest:** The authors declare no conflict of interest.

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
