# Peer review of "Treatment of Scrap Tire for Rubber and Carbon Black Recovery"

_recycling, doi:10.3390/recycling7030027_

Round 1

Reviewer 1 Report

There are still numerous places with grammatical errors. Suggest to have a native speaker to proof read this manuscript.

Overall, my previous comments are addressed sufficiently. 

Author Response

Dear reviewer 1:

Thanks for you're reviewing the manuscript titled

"Tires Scrap Treatment for Rubber and Carbon Black Recovery"

*Almost all your comments were done.

* The English language has been improved.

Reviewer 2 Report

Recycling scrap tires is an interesting research subject and could be realized in three ways: mechanical recycling into small pieces for direct use, combustion as fuels, or decomposition to chemical constituents. The submitted work is focused on the chemical recycling of tires into rubber and carbon black. This route presents some novelty in the chemical recycling of tire waste because, in general, the isolation of low molecular weight products was reported. Despite the new idea, I did not find any comparison with the previously reported chemical recycling method.

In my judgment, the submitted manuscript is more like a patent or industrial report than a scientific article. I recommend the publication after a major revision.

The list of my suggestions and questions is presented below:

Line 59:

“Tires scrap of trucks from local market were collected and prepared for treatment by cutting them into pieces which then cleaned and grinded.”

Dimensions of the pieces used should be given.

Line 72: “The soaked solution was poured using a 250ml round bottom flask. 0.5g of 4-Hydroxy TEMPO catalyst has been added with 1.0g of alumina as catalyst support. The flask contents were heated at the range of the boiling point of gas oil (250-350) °C. After two hours of heating the contents, the process was stopped, leaves to cool down, then filtered to get a clear black solution.”

4-hydroxy TEMPO is an organic compound that is well soluble in gas oil, while Al2O3 is insoluble in inorganic/organic solvents, and therefore Al2O3 cannot act as catalyst support. In heterogeneous catalysis, increasing the surface area involves distributing the catalyst over the surface of the support. However, here there is no heterogenized catalysis.

What is the role of 4-Hydroxy TEMPO and Al2O3 during tires chemical recycling? Tires scrap recycling was carried out between 250-300 C and is 4-hydroxy TEMPO stable under these conditions.

The authors should perform three additional reactions for the recycling of tires scrap: a) without any catalyst source, b) in the presence of 4-hydroxy TEMPO, and c) in the presence of Al2O3. Based on the results obtained, the action of each species should be explained.

Line 55: “In addition, 4-Hydroxy TEMPO catalyst was used to help rubber chains dissolve faster in gas oil.”

and

Line 367: “The activity of gas oil solvent has shown elevation in the presence of the 4-Hydroxy-TEMPO catalyst which is used as a polymerization inhibitor for the rubber at high temperature in the reflux process.”

The authors state that 4-hydroxy-TEMPO is a rubber polymerization inhibitor, so what is its catalytic role.

Line 153: The authors should analyze the FTIR spectrum and introduce information about the C-S and S-S bands because vulcanized rubber was used as the starting material.

Line 156: I suggest the change of paraffinic protons for alkyl protons.

Line 161: Figure 2 is not very informative because it contains high-intensity peaks of HDO and DMSO-D6. Signals corresponding to the polymer material should be more exposed.

Lines 173, 289, 319: The PXRD pattern revealed some amorphous phases in the samples analyzed. In Figure 3, the amorphous was carbon black, organic oils, and polymeric materials from tires. Carbon black is usually identified as the basis on the broad diffraction peaks at 20-30 and 45 2theta, but the authors do not give any information about its identification in the starting material. The PXRD presented should be analyzed using powder diffraction databases, i.e., PDF4+ or Crystal Open Database, to identify the crystalline component of the study material.

Lines 179 and 317: Table 1 and Table 4 should be removed to SI.

Line 213: During the SEM study, the authors did not perform an EDS analysis to obtain information about the metal elements presented in the sample. Metal oxides, i.e., ZnO, were used very often for tires production.

Line 247: “Even the 4-Hydroxy-TEMPO catalyst was found more active among many others catalysts used for the same purpose.”

However, I did not find any information on other catalysts used for the same purpose.

Lines 251-259: 3.2.1. FTIR study.

The FTIR analysis did not contain information on the C-S and S-S bands. Both bands were present in the resulting spectrum.

Line 271: Figure 8. 1H NMR spectrum of recovered rubber.

1H NMR analysis revealed the presence of various polymers in the isolated sample. The spectrum presented is complex but contains all the necessary information. For clarity, only the most intensive peaks should be labeled, e.g., 7.25 for styrene groups, etc. The integration labeled should be removed from the 1H NMR spectrum because it is difficult to say what it means. This spectrum should show only the region from 0 to 9 ppm.

Line 346: “The SEM images show the pyrolytic CB particles are spherical with nanosize, and their whitening surface indicates that they have a high crystalline structure.”

From the SEM picture presented, it is not easy to deduce that the investigated material is highly crystalline. For SEM studies of semicrystalline materials, surface whitening is also usually observed.

Lines 363-377: The “chemical dissolution process” reported by the authors used to recover rubber and carbon black from truck tires is a method, in my opinion, based on partial chemical degradation of the starting material caused by the action of high temperature, the solvent used, and 4-hydroxy-TEMPO and Al2O3. Finally, this method separated rubber and carbon black, but it was not a simple dissolution process. I do not find any chemical explanation for the reactions performed. During this process, did the C-S and S-S bonds remain intact? What is the Mw of the isolated rubbers relative to the Mw values in the starting material? What is the sulfur content of the isolated rubbers relative to the sulfur content of the starting material?

Author Response

Dear Reviewer 2:

Thanks for all the points you asked to be done or those need discussion in your revision of the manuscript titled" Tires Scrap Treatment for Rubber and Carbon Black Recovery" and the following points I need to explain:

1- Most of the previous recycling processes of waste tires are couldn't find way for comparison with our work because they concentrate on modulation of tires into fuel oil or grinded and used as it is. Furthermore our recovered rubber wasn't low molecular weight product (the different analysis were improved that).      

2- The submitted work isn't industrial report but it is a PhD thesis of my research scholar              Miss Alaa Sultan and we have the intention to write other parts of this work as patent.  

3- What was mentioned in your question in the Line 59: was done in the manuscript.

4- What was mentioned in your question in the Line 72 : The used catalyst has boiling point           of 269oC and it play very good role in the homogeneity of rubber/gas oil solution in                 comparison with non-catalytic reaction especially  before reaching the boiling point of              gas oil. In addition the catalyst play good role in inhibiting the rubber polymerization at            height temperature degrees.

5- What was mentioned in your question in the Line 55 and line 367: 4-Hydroxy TEMPO was      used as catalyst for two purposes first increase the homogeneity of the rubber solution               which is very important for penetration of the solvent molecules between rubber chains and      surrounding them. Second as polymerization inhibitor of the rubber polymer and this is             considered the best properties of the used catalyst ( a reference (No.24) was given in the           manuscript for that).    

6- What was mentioned in your question in the Line 153: was done in the manuscript.

7- What was mentioned in your question in the Line 156: was done in the manuscript.

8- What was mentioned in your question in the Line 161: The position of the 1H NMR signals       are clear as native tire which is not pure compound.

9- What was mentioned in your question in the Line 173, 289, 319: was done in the                        manuscript except the powder diffraction database because it is not available and the                 crystalline structure of the studied materials are clear and was explained well.

10- What was mentioned in your question in the Line 179 and 317: was treated in the                      manuscript.

11- What was mentioned in your question in the Line 213: EDS analysis for metal elements is        not available in our SEM instrument.

12- What was mentioned in your question in the Line 247: was treated in the manuscript( the         sentence was taken direct from the thesis which was used five catalysts without data                  processing.

13- What was mentioned in your question in the Line 251-259: was done in the manuscript.

14- What was mentioned in your question in the Line 271: was done in the manuscript.

15- What was mentioned in your question in the Line 346: was done in the manuscript.

16- What was mentioned in your question in the Line 363-377: I would like to say the rubber from the waste tires was recovered by chemical dissolution process for the first time and not through partial chemical degradation. Recently we done the process under height pressure and in a low temperature in comparison with previous procedure and it gives good results. The molecular weight values are not easy done because tires have many types of rubbers and it is not easy to make combinations between.

Please see the resubmit manuscript and all the revised points were marked in yellow.

Thanks

Dr.Fawzi Habeeb Jabrail

Round 2

Reviewer 2 Report

Thank you very much for the attached manuscript. I found some positive responses to the previous suggestions. However, my negative objective here is involved with the catalytic action of TEMPO, and in my opinion, it is not a catalyst but an antypolymerization agent. The Author response: “Even the 4-Hydroxy-TEMPO catalyst was found active in comparison with the non-catalytic process in recovering the rubber and CB from tire, and this is clear from the yield% of the recovered rubber from the non-catalytic process of 34.4% in comparison with 38.2% for the catalytic process.” confirmed my opinion. What is the role of Al2O3? It is not catalyst support. For example, the catalyst support (material containing immobilized catalyst) could be MgCl2 during olefin polymerization, but not Al2O3 here. I have many questions before any answers or comments.

Author Response

Dear Reviewer 2:

 For your question "my negative objective here is involved with the catalytic action of TEMPO, and in my opinion, it is not a catalyst but an anti-polymerization agent"

My response is (( I am satisfy that the used 4-Hydroxy TEMPO is a catalyst and it works in two directions, as polymerization inhibitor and facilitates gas oil molecules in their penetration process between the rubber chains. The recent work done on other catalysts like TiO2 shows almost same rubber yield percentage but give low molecular weight polymer. In addition the Al2O3   was working as supporter for TEMPO and in its absence both catalysts TEMPO and TiO2 they don’t play any role in the recovery of rubber from tires.

Thanks for reviewing our manuscript,

This manuscript is a resubmission of an earlier submission. The following is a list of the peer review reports and author responses from that submission.

Round 1

Reviewer 1 Report

The subject of the paper clearly falls within the scope of this Journal.

The paper is very interesting, well written and well organized, and represents some advancement over the actual state-of-the-art. The ways and means are well described as well as the obtained results which are thoroughly discussed and conclusions are well drawn. The paper is also supported by some literature review. However, in order to make for a stronger paper, I suggest that the authors should cite and discuss the following relevant paper:

- Gomes et al., “Toxicological Assessment of Coated versus Uncoated Rubber Granulates Obtained from Used Tires Aiming to its Use in Sport Facilities”, Journal of Air and Waste Management Association, 60(6), 741-746 (2010)

DOI: 10.3155/1047-3289.60.2.1

I do recommend the publication of this paper, subjected to these changes.

Reviewer 2 Report

Overall, this manuscript just reads like a test report without necessary discussion/rationale on the major findings of the study.

Presentation quality of figures and test results are poor. Also lack of the necessary detail for readers to fully understand author's analysis and experiments.

The researcher should also sufficiently interpret the reasons of the research outcome, which will support future research.

Please see attached for other comments
